# Exogenous FGF-1 Differently Regulates Oligodendrocyte Replenishment in an SCI Repair Model and Cultured Cells

**DOI:** 10.3390/biomedicines10112724

**Published:** 2022-10-27

**Authors:** Meng-Jen Lee, May-Jywan Tsai, Wen-Chi Chang, Wang-Yu Hsu, Chien-Hui Hung, Ya-Tzu Chen, Tsung-Hsi Tu, Chih-Hung Shu, Ching-Jung Chen, Henrich Cheng

**Affiliations:** 1Department of Applied Chemistry, Chaoyang University of Technology, Taichung 41349, Taiwan; 2Neural Regeneration Laboratory, Department of Neurosurgery, Neurological Institute, Veterans General Hospital, Taipei 11217, Taiwan; 3Division of Neural Regeneration and Repair, Neurological Institute, Taipei Veterans General Hospital, Taipei 11217, Taiwan; 4Faculty of Medicine, National Yang Ming Chiao Tung University, Taipei 11221, Taiwan; 5Department of Otolaryngology, Taipei Veterans General Hospital, National Yang Ming Chiao Tung University, Taipei 11217, Taiwan; 6Institute of Pharmacology, National Yang Ming Chiao Tung University, Taipei 11221, Taiwan

**Keywords:** Fibroblast Growth Factor (FGF)-1, spinal cord injury, oligodendrocyte progenitor cell (OPC), NG2, demyelination

## Abstract

We studied the phenotypes in an oligodendrocyte genesis site at the acute stage of spinal cord injury, when we observed regenerated ascending neurites. Pan-oligodendrocyte marker OLIG2+ cells were more in fibroblast growth factor (FGF)-1-treated rats (F group) than in non-treated (T group) in this site, while the number of NG2+OX42− oligodendrocyte progenitor cell (OPC), CNPase+ OPC, Nkx2.2+ OPC, and APC+ remyelinating oligodendrocytes was less in the F group. Paradoxically, when we label the rats with pulsed bromodeoxyuridine (BrdU), we found that the mitotic NKX2.2+ OPC cells are more in the F group than in the T group. We tested the embryonic spinal cord mixed culture. FGF treatment resulted in more NG2(+) CNPase (+) than non-FGF-1-treated culture, while the more mature NG2(−) CNPase(+) cell numbers were reduced. When we block the FGF receptor in the injured rat model, the NG2+OX42− cell numbers were increased to be comparable to non-FGF-1 rats, while this failed to bring back the APC+ mature oligodendrocyte cell numbers. As migration of OPC toward injury is a major factor that was absent from the cell culture, we tested 8 mm away from the injury center, and found there were more NG2+ cells with FGF-1 treatment. We proposed that it was possibly a combination of migration and proliferation that resulted in a reduction in the NG2+ OPC population at the oligodendrocyte genesis site when FGF-1 was added to the spinal cord injury in vivo.

## 1. Introduction

Transection of the spinal cord induces Wallerian degeneration of the distal stumps, and apoptosis occurs in the proximal stumps via macrophage invasion and inflammation-mediated degeneration. This degeneration will be evolved to form a white matter cavity and a gliosis reaction occurs to seal this cavity, sparing the residual stumps. The molecular and cellular organization of the gliosis scar, residual tissue, and rim of the cavity is very important to general functional recovery during a spinal cord injury (SCI) because this region contains surviving axons, and molecular contents of this region greatly influence axonal outgrowth.

In axon regeneration, different myelin components may serve opposing functions, which may depend on the integrity of the myelin and the relationship shared between myelin and axons [1,2,3]. Therefore, two contradictory actions must occur in the residual stumps during SCI recovery: (1) contact with myelin proteins must be prevented prior to axon regeneration, and (2) myelination and OPC proliferation must occur after axon regeneration. To achieve this delicate balance, more information is needed to fully understand oligodendrocyte physiology in the residual stump during the acute phase of SCI. The glial scar is typically formed in tissue adjoining the lesion cavity and is a dynamic region in which post-SCI oligoprogenitor division and oligodendrogenesis occur [4,5,6]. New NG2+ cells generally get accumulated in glial scar and co-localize with astrocytes, microglia, and macrophages [7,8,9]. Continued oligodendrocytic apoptosis and post-injury oligodendrogenesis result in restored or increased numbers of oligodendrocytes in the spared white matter [5,10]. Many signals trigger oligodendrocyte proliferation, maturation, and survival. Oligodendrocytic death may be induced by axotomy [11], excitotoxicity [12], microglial activation, inflammatory cytokines, oxidative stress [13] demyelination [14]; or P75 nerve growth factor receptor (NGFR) activation by pro-NGF and the death receptor Fas [15,16]. The proliferation of oligodendrocyte progenitors is regulated by many factors, including progesterone [17], bone morphogenetic protein 4 (BMP4) [18], Platelet-derived growth factor A (PDGF-A), insulin-like growth factor 1 (IGF-1), Ciliary neurotrophic factor (CNTF), Neurotrophin-3 (NT-3), brain-derived neurotrophic factor (BDNF), chemokine (C-X-C motif) ligand 1 (CXCL-1) [19], and FGF-2. In particular, FGF-2 may stimulate oligodendrocyte progenitor proliferation via activation of FGF receptor (FGFR) 1; however, FGF-2 may also induce the loss of mature oligodendrocytes and myelin [20,21,22].

One of the SCI treatments involves the usage of peripheral nerves and FGF-1 in combination to regenerate axons of the transected spinal cord [23]. At 28 days post-operation, in the back-degenerative sites of the ascending dorsal funiculus tracts, the number of phagocytosing macrophages and the size of the white matter cavity were reduced [24]. Additionally, arginase-I and M2 macrophages were predominantly observed in the grafted nerves and the injury site at 10–14 days post-operation [25], and neurite outgrowth was observed entering the nerve graft as early as 4–7 days post-operation [24]. Our previous treatment of FGF-1 and peripheral graft has rendered the complete spinal transected rats with partial functional recovery, which includes the Basso, Beattie, and Bresnahan (BBB) score that was significantly improved; detection of somatosensory (SSEP) and motor evoked potentials (MEPS) showed both sensory and motor information across the damaged site was improved [26]. Regeneration of axons throughout the graft and into the distal stump was demonstrated by retrograde labeling in the cortex and brain stem nuclei and anterograde labeling occurred in the distal stump [27]. The choice of FGF-1 was empirical, although FGF-2 was similar but much better explored and is a known factor.

The mechanisms by which our combined treatment of FGF-1 and peripheral graft acts to enhance axon regeneration remain unclear. FGF-1 was previously reported to activate the same repertoire of receptors as observed with FGF-2 [28], that reverses mature oligodendrocyte phenotypes and increases the number of oligodendrocyte progenitor levels [20,21]. In this study, we would like to investigate whether exogenous FGF-1 affects OPC and oligodendrocyte physiology significantly, in SCI. Comparing untreated and treated transected rats provides an opportunity to examine the mechanism by which the specific aspects of degenerative events affect oligodendrocyte physiology.

We used several markers of oligodendrogenesis to study the progression of OPC to mature oligodendrocyte when the complete transected spinal cord is treated with a therapy containing FGF-1. During development, OLIG2 promotes the formation of oligodendrocyte precursors and differentiation, and was expressed in motor neuron progenitors, neural stem cells, OPC, and throughout cells of oligodendrocyte lineage [29,30]. NG2+ is a marker for the oligodendrocyte progenitors during development and in oligodendrogenesis during CNS nerve injury. Almost 30% of pericytes in the wound area are NG2+ cells and are important for the generation of fibroblastic scar [31]. NG2+ glia does not enter fibrotic lesions [32], but they accumulate within the glial scars in rodents and humans. NG2 is also present in stem/progenitor cells within the peripheral nerves [33]. NG2 was reported as a marker for OPC, and although also expressed by pericyte and stem/progenitor of other tissue [34,35,36], it is expressed by OPC in normal spinal cord, and was elevated during spinal cord injury [4,5,37,38,39]. The Nkx2.2 is a key regulatory gene that initiates oligodendrocyte differentiation. It is up-regulated in OPCs during the differentiation stage [40], and down-regulated when OPC proliferation stops and becomes pre-myelinating oligodendrocytes [29]. Co-expression of Nkx2.2 and OLIG2 in embryonic chick spinal cord promotes ectopic expression of myelin genes [29,41,42]. CNPase is expressed exclusively by oligodendrocytes in the CNS, and its appearance is among the earliest events of myelinating oligodendrocyte differentiation [43]. The sequential expression of these markers, although oversimplified, is listed in Table 1 for convenience of understanding the rationale of the experimental design.

## 2. Materials and Methods

### 2.1. Animals

Adult female Sprague-Dawley rats that weighed 200–250 g were used in this study. The animals were provided by the animal facility of National Yangming University, Taipei, Taiwan. After the transection operation, animals were kept in individual cages located in ventilated, humidity- and temperature-controlled rooms with a 12/12 h light/dark cycle. They received food pellets and water ad libitum. 

### 2.2. Surgical Procedures

All procedures involving animals were approved by the Institutional Animal Care and Use Committee of Taipei Veterans General Hospital (IACUC approval number: 2019-073, approval date: 27 June 2019 and approval number: 2020-172, approval date: 3 September 2020). The following operation procedures have been previously described in detail elsewhere [44]. Briefly, operations were performed on a heating pad under general isoflurane anesthesia. Anesthesia was induced with 5% isoflurane in air and maintained with 1–2% isoflurane in air via a VapomaticTM anesthetic vaporizer (A.M. Bickford Inc., Wales Center, NY, USA). Rectal temperatures were monitored during surgery and maintained at no less than 35 °C. Antibiotics (Borgal; Yung Shin pharm. Ind. Co. Ltd., Taichung, Taiwan) were administered (42 mg) immediately before surgery and daily. To avoid urinary tract infections, manual emptying of the urinary bladder was performed 3 times daily during the first week and twice daily thereafter.

### 2.3. Animal Treatment Groups

For the rats that underwent the transection operation without treatment (T group), their spinal cords were completely transected and 5 mm of the T8 spinal cord segment was removed. For rats that received the combined treatment immediately following the transection operation (R group), 12–18 autologous intercostal nerves were collected from the same rat and implanted with FGF-1 mixed in fibrin glue to seal the spinal segment area. Post-operative care and monitoring were performed as previously described [44]. Some rats received transection operation followed by FGF-1 supplemented fibrin glue filled in the transected area (F group) Spinal cords were collected 4–7 days post-operation (P.O.) for most data except for the neurite outgrowth 14 P.O. samples were included. A 2 cm segment of the spinal cord of each rat, including the middle scar or nerve graft, was processed by cryosection and immunohistochemistry. The authors thank Eusol Biotech Co., Ltd., (Taipei, Taiwan), for providing FGF-1.

### 2.4. FGF-R Blocker

In the ‘blocker’ experiments, the glue in rats of R and F groups was additionally mixed with FGF-R1 blocker PD173074 at 200 μM concentration. The number of NG2+Ox42− cells was counted in the back-degenerative dorsal funiculus tract (centered around 6 mm from the trauma center). This is used to test whether blocking of FGF action would result in alteration of OPC generation.

### 2.5. BrdU Injection

The operated rats were injected intraperitoneally with 200 mg/kg bromodeoxyuridine (BrdU; Sigma, St. Louis, MO, USA) in one of the two different experimental paradigms. A short terminal BrdU pulse (injections at 4 h and 2 h before sacrifice) [45] was used to detect actively cycling cells without an intervening period for subsequent differentiation to occur. A long-term BrdU administration protocol [46] was used to analyze the total proliferative response at 1–4 days P.O. Each rat was injected twice (12 h between injections) on 1 and 3 days P.O. BrdU injections were given on alternate days to avoid concerns of potentially harmful effects from high levels of BrdU incorporation. Immunohistochemistry combined with BrdU detection was carried out as follows [45]. After histochemistry, sections were treated with HCl and then incubated overnight with a monoclonal anti-BrdU antibody directly conjugated with horseradish peroxidase (diluted 1:15; Boehringer Mannheim). Peroxidase activity was detected by incubation with 3,3-diaminobenzidine (DAB; Vector Laboratories, Burlingame, CA, USA).

### 2.6. Tissue Preparation

Rats were anesthetized with pentobarbital and perfused with 4% paraformaldehyde (PF) in 0.1% phosphate-buffered saline (PBS). The spinal cord was collected and fixed overnight in 4% PF in PBS. The tissue samples were washed with PBS several times, 1 h each, and then immersed in 15% and 30% sucrose in PBS for cryoprotection. Tissue blocks were embedded in Jung tissue freezing medium (Leica, Wetzlar, Germany) and frozen in liquid nitrogen-cooled 2-methylbutane (−80 °C). Spinal cords were transversely sectioned at 20 µM on a cryostat. Cross-sections were systematically labeled, and distances from trauma centers were calculated and noted. The trauma center was determined as the midpoint of two graft–stump or scar–stump junctions. The cross-sections in this study were taken from the dorsal column that was caudal to the transection site; therefore, they represent the proximal stumps as opposed to the Wallerian stumps of mostly sensory tracts (see also Figure 1).

### 2.7. Immunohistochemistry

Spinal cord sections were incubated with the antibodies of interest. The sections were then washed in PBS and incubated with secondary antibodies that were conjugated with fluorophores, or secondary antibodies that were coupled with biotin and streptavidin-conjugated fluorophores, to increase sensitivity. For double immunostaining, the secondary antibodies used were Alexa 488 fluorophore donkey anti-rabbit antibody (1:200, Molecular Probes, Waltham, MA, USA) and Cy3-conjugated donkey anti-mouse antibody (1:200, Jackson ImmunoResearch Laboratories, West Grove, PA, USA). Primary antibody omission controls were performed for all immunostaining protocols to control for non-specific binding. Fluorescent visualization and photography were performed using a Zeiss Axioscope microscope with the appropriate filters. GM hematoxylin and eosin (HE) solution was purchased from Muto Pure Chemical Co., Ltd. (Tokyo, Japan). After primary antibody incubation, the sections were incubated in ABC mixture (Vector Labs, Newark, CA, USA) to subsequently develop peroxidase via incubation in DAB solution to visualize the peroxidase label, or by using a chromogen development kit that was purchased from Vector Labs.

The antibodies that were used for these procedures were as follows: CC1 (anti-APC, 1:20; mouse mono-clone, Millipore, Darmstadt, Germany, cat NO:OP80), NG2 (1:200; polyclonal, affinity purified, Millipore, cat NO: AB5320), OX42 (mouse monoclonal, 1:100, Bio-Rad, Hercules, CA, USA, cat NO: MCA275R), NOGO A (goat polyclonal, 1:500, Santa Cruz Biotechnology, Santa Cruz, CA, USA, cat NO: sc-11032), OLIG2 (rabbit polyclonal anti-Olig-2, 1:500; Millipore, cat NO: AB9610), Nkx2.2 (H-60, Rabbit polyclonal, 1:100, Santa Cruz sc25404), Caspase-3 (rabbit polyclonal 1:200; Cell Signaling Technology, Danvers, MA, USA, cat NO: #9662), RIP (anti-CNPase, 1:500, mouse monoclonal, Developmental Studies Hybridoma Bank (DSHB), Iowa City, IA, USA), 15A3 (DNA and RNA oxidative damage markers, mouse monoclonal, 1:100; QED Biotechnology, San Diego, CA, USA, cat NO: 12501), anti-fibronectin (rabbit polyclonal, 1:500, Dako, Carpinteria, CA, USA, cat NO: A 0245), and anti-beta III tubulin (mouse monoclonal, 1:500, BioLegend, San Diego, CA, USA, cat NO: MMS-435P), Ki67 (for proliferating cells, 1:500; Abcam, Cambridge, MA, USA), BrdU (1:200; Chemicon International, Temecula, CA, USA), GFAP (astrocytic marker, Dako, Carpinteria, CA, USA). These antisera were purchased from a well-established biotech company, and the antigen was disclosed, specificity tested, and the staining patterns were well known in the species and tissue in which the experiment was performed. All immunostainings were accompanied by a control test in which the primary antiserum was omitted. Photographic images were acquired for quantitative comparison using the same settings for each experiment. For individual analysis methods, please see the following. Tunnel staining was performed using DeadEnd™ colorimetric TUNEL System according to the manufacturer’s instructions (Promega corporation, Madison, WI, USA).

### 2.8. Area of Interest

Briefly, a 0.5 cm segment was removed from the spinal cord by two incisions flanking the segment followed by lifting the enwrapping dura and complete removal of the segment. The correspondent distance from the trauma center, to the incision site, was therefore 2.5 mm rostrally and caudally (Figure 1, part B). After immunostaining and suitable photography, data were analyzed in the caudal stump of the dorsal column in sections located 5–7 mm from the trauma center (Figure 1, part C), where we have consistently detected reactive gliosis previously [44].

### 2.9. The Sampling Methods for OPC, Maturing Oligodendrocytes, and Apoptotic Oligodendrocytes

The rostral dorsal column was selected for examining the back-degenerative sensory tracks, for the cavity consistently appeared in the middle of the dorsal column. The gliosis site could therefore be double-checked with their distance from the trauma center, as well as from the cavity-forming anatomical features, especially for the transected without treatment group (see also Figure 1). The cell numbers were analyzed by analyzing the digital double- or triple-labeled photographs that were merged from the respective channels of the fluorescent pictures. For analyzing NG2/Ox42− immature progenitors, NOGO+/Oligo2+ maturing oligodendrocytes, and APC+/caspase-3+ for apoptotic oligodendrocytes, 100× photographs were taken, and the total number of positive cells within 600 × 600 pixels (2.5 × 10^5^ μm^2^) square selected from the middle area of the rostral dorsal column was counted.

### 2.10. Staining and Analysis for Nucleic Acid Oxidation

Using an ABC kit to develop chromophores, cross-sections were immunostained with antibody against ubiquitin, which is a marker for axonal degeneration, and 15A3, which is a marker for nucleic acid oxidation. The sections were photographed under equal lighting, and digital images were captured from serial sections for analysis. 100× photographs were taken, and the IR within 600 × 600 pixels (2.5 × 10^5^ μm^2^) selected from the middle area of the caudal dorsal column were analyzed. The number of pixels analyzed via ImageJ to have values ranging from 0–100 was calculated and used to estimate IR.

### 2.11. Staining and Analysis for Myelin

Cross-sections were stained with Luxol fast blue to visualize myelin following standard protocols. Proportions of the demyelinated areas were manually estimated as either 25, 50, 75, or 100% and averaged across samples.

### 2.12. Staining for the Neurite Outgrowth

Cross-sections were collected from the grafted nerves. Antibody for fibronectin was used to stain the epineurium of the nerves, and TUJ-1 antibody stained both the remaining axons as well as the newly formed neurites. The remaining axons of the nerve graft and the new neurites could be distinguished by their locations; the remaining, degenerating axons are encircled by the fibronectin-rich epineurium, and the newly formed neurites grew on the fibronectin-rich epineurium.

### 2.13. Cell Culture

Mixed neuronal/glial cell cultures were prepared from the spinal cord regions of embryonic SD rat fetus at gestation 14–16 days as described in our published articles [47]. Briefly, cells were dissociated with mixtures of papain/protease/deoxyribonuclease I (0.1%:0.1%:0.03%) and plated onto poly-D-lysine coated dishes at a density of 1–2 × 10^5^ cells/cm^2^. The cells were maintained in Dulbecco’s Modified Eagle’s Medium (DMEM) supplemented with 10% FBS and incubated at 37 °C in a water-saturated atmosphere of 5% CO_2_/95% air. Cultures were treated with aFGF (100 ng/mL) or bFGF (50 ng/mL) on the second day after cell seeding in DMEM supplemented with N2 ((Invitrogen, for serum-free condition). FGF-2 was purchased from peproTech (Rocky Hill, NJ, USA). The authors thank Eusol Biotech Co., Ltd., (Taipei, Taiwan), for providing FGF-1. Cultures were refilled with medium and factors on the third day after treatment. At the end of the experiment, cultures were fixed with 4% paraformaldehyde and processed for immunohistochemistry. Images of immunoreactive (IR) cells were obtained under a fluorescent microscope equipped with a cooling digital imaging system. Beta III tubulin-IR neurite density, NG2-or Rip-IR cells (obtained using a 20× object lens) were analyzed using Image-Pro Plus software (NIH systems, Bethesda, MD, USA). A one-way ANOVA was performed; differences were considered to be statistically significant when *p* < 0.05 (*).

## 3. Results

### 3.1. Treatment of FGF-1 to the Spinal Cord Transection Model Explained, and Demonstration of Axonal Ingrowth into the Nerve Graft

#### 3.1.1. Area of Interest

Briefly, a 0.5 cm segment was removed from the spinal cord by two incisions flanking the segment followed by lifting the enwrapping dura and complete removal of the segment. The correspondent distance from the trauma center (A), to the incision site (B), was therefore 2.5 mm rostrally and caudally. After immunostaining and suitable photography, data were analyzed in the caudal stump of the dorsal column in sections located 5–7 mm from the trauma center (C), where we have consistently detected reactive gliosis previously [44].

#### 3.1.2. TUJ-1 Positive Axonal Outgrowth in the Grafted Nerves

After our spinal cord repair procedure, the axons were seen to grow and reach the grafted nerves. Cross-section of the nerves demonstrated that the original PNS axons were undergoing degeneration. Some TUJ-1-positive neurites were seen closely associated with the epineurium in nerves that were devoid of the central axons (Figure 2a). The axons were significantly devoid of their original central axons and epineurium filled with more neurites 14 days after injury (Figure 2b).

### 3.2. Differentiation of Cells of the Oligo Lineage upon FGF-1 Treatment in the Injury Prenumbra

#### 3.2.1. OLIG2+ Cells Were Increased in R4 and F4 Rats

OLIG2 is a pan-oligodendrocyte marker that stains throughout relatively early OPC to mature oligodendrocytes [48]. An increase in OLIG2+ cells was found in our area of interest (6.24 mm from the trauma site) in both the R4 and F4 models, demonstrating that this effect of FGF-1 on the oligodendrocytes in the injury site was significant with or without the nerve graft (Figure 3).

#### 3.2.2. NG2+OX42− OPC Were Decreased in R4 and F4

Injury to CNS axons has been shown to stimulate the proliferation and maturation of OPCs, as well as the resulting replacement of the OL population [49]. Our transection model resulted in similar outcomes; in the proximal stump, NG2+OX42− OPCs were present in the injury penumbra (Figure 4). NG2 is a marker of OPC, and its IR was present at the prominent site of oligodendrogenesis following CNS axotomy [37,50]. Unfortunately, NG2 also labeled microglia and macrophages [38]. We used staining for Ox42 to exclude microglia and microglia-derived cells. The number of NG2+Ox42− OPCs was significantly different between the T4 and R4, as well as between T4 and F4 rats at 6–6.24 mm from the trauma center (Figure 4).

#### 3.2.3. Analyzing the Effect of FGF-1 to the Injury Site: Use of F Model and Verification of NG2+ Cells as True Oligo Progenitor Cell

Some functional recovery was observed in the R model. However, it was a very time consuming and complicated operation and involved autografts from the same patient. It would be advantageous when the therapy would be translated into the clinical setting if the R model could be simplified in any way. In fact, contusion resembles the everyday situation much better, and the application of factors alone is more feasible than using any nerve graft. In the following, we used the F model instead of the R model to test the effect of the singular FGF-1 application on the transected spinal cord. The identification of the NG2+ cells was further verified around the injury site. NG2+OX42− OPC at the oligodendrogenesis site are morphologically different from the NG2+OX42+ cells, as the morphology of the latter was rounded while the former was slender with branches. Those slender-branched NG2+ were mostly stained with OLIG2, demonstrating that they were bona fide OPC. The number of OLIG2+NG2+ cells in this specific area was analyzed and was reduced in the F4 compared to T4 (Figure 5).

#### 3.2.4. FGF-1 Treatment Results in Less CNPase+NG2+ and Less CNPase+NG2– Cells at 4 Days Post Operation

CNPase is expressed exclusively by oligodendrocytes in the CNS, and is one of the earliest events of myelinating oligodendrocyte differentiation [51]. As the CNPase+ population matures, they gradually differentiate from OPC to become early myelinating oligodendrocytes and lose the NG2 marker. The comparative evolution of CNPase/NG2 markers delineates the degree of maturation. When the transected spinal cord was treated with FGF-1, both the CNPase+NG2+ and CNPase+NG2− populations were reduced compared to the non-treated transected rats (Figure 6). This also resulted in a reduction in total CNPase+ cells.

#### 3.2.5. Nkx2.2+ OPC Were Reduced at Specific Site in R4

When examining the oligodendrocyte progenitors by staining of Nkx2.2, the cell numbers that were present in the oligodendrocyte genesis site were less in the FGF-1 added cells when compared to the transected rats (Figure 7).

#### 3.2.6. APC+ Mature Oligodendrocytes Were Reduced at Specific Site in R4

When examining the mature oligodendrocytes by staining the APC, the cell numbers that were nearest to the open wound were less in the FGF-1-added cells when compared to the transected rats (Figure 8). As stated in the following, cell death was not significantly different between the two models; therefore, the fact that mature oligodendrocytes, which stained APC cells, were less in the FGF-1-treated rats nearer to trauma suggested a delayed oligodendrogenesis in these rats.

### 3.3. Cells of the Oligo Lineage Proliferate and Maintain an Early Phenotype upon FGF-1 Treatment in the Injury Prenumbra, while the Apoptosis Rates Were Similar

#### 3.3.1. Cells of the Oligodendrocyte Lineage Proliferate upon FGF-1 Treatment to Generate More Nkx2.2-Positive Cells in the FGF-1-Treated Rats

Nkx2.2 is expressed in the OPC and down-regulated as the cells become the pre-myelinating oligodendrocytes [29,52]. When the rats were injected with BrdU daily after operation and the sample was collected at 4 days P.O., we found that cells doubled stained with BrdU and Nkx2.2 were significantly increased in our FGF-1-treated rats (Figure 9, long pulse). However, when the rats were injected 3 h before sacrifice, the proliferation was not significant, suggesting that the cells do not proliferate as much 4 days post operation (Figure 9, short pulse).

These data corroborated with others finding that FGF was mitogenic for cells of oligo lineage [22], and also indicated that the reduction in the NG2(+) CNPase(+) population in FGF-1-treated rats at 4 days P.O. was not due to accelerated maturation, but maintenance of phenotypes that were earlier than the maturation stage.

#### 3.3.2. Number of Apoptotic Oligodendrocytes Was Not Different upon FGF-1 Treatment

The apoptosis rate of the injury penumbra was examined. The general apoptotic cell numbers demonstrated by the tunnel staining (Figure 10), were not different between the transected rats (T model) and the FGF-1-treated rats (R model). This demonstrated that the reduction in the mature oligodendrocyte population was not due to the preferential death of oligodendrocytic cells, but a reduced rate of producing them.

The apoptosis rate of the mature oligodendrocytes was examined. The caspase3+ APC+ cells (Figure 11), which represented the dying mature oligodendrocytes, were not different between the transected rats (T model) and the FGF-1-treated rats (R model). This demonstrated that the reduction in the mature oligodendrocyte population was not due to the preferential death of oligodendrocytic cells, but a reduced rate of producing them.

### 3.4. Myelin Integrity: Rats Treated with FGF-1 Underwent Significantly Less Demyelination in the Dorsal Column at Site of Reactive Gliosis

Luxol fast blue staining demonstrated that demyelination was significantly greater in the center of the dorsal column of the T4 rats. Area of significantly less demyelination was located at the periphery of the developing cavity and coincided with the oligodendrocyte-generating zone as stained with NG2. Myelin was generally more well preserved in the rats that received the treatment containing FGF-1, which includes the R model and F model (Figure 12).

### 3.5. Mixed Neuron-Glia Culture Yields More Cells of Immature Phenotypes When FGF-1 or FGF-2 Were Added in Short Term Culture

The cells were cultured with FGF-1 (100 ng/mL) or FGF-2 (50 ng/mL), and were fixed and double stained with NG2 and CNPase at 2 and 4 days after FGF treatment. The number of 3 groups of cells was compared: NG2(+) CNPase(−), NG2(+) CNPase(+) and NG2 (−) CNPase (+). NG2 was expressed by oligodendrocyte precursor (OPC), and CNPase expression was a more mature phenotype than NG2 as the OPC differentiate into myelinating oligodendrocyte [51]. The samples that receive FGF treatment have slightly more NG2(+) CNPase(−) cells than untreated ones and are observed to be the OPCs. When the CNPase(+) populations were analyzed, FGF treatment resulted in more NG2(+)CNPase (+) than untreated ones, while the more mature phenotype NG2(−)CNPase(+) was reduced in the FGF-treated samples. This phenomenon was the same in the 2nd and 4th days P.O., with the difference in the NG2(+)CNPase(+) population more prominent in the 4-day culture (Figure 13). This observation was corroborated by other reports that FGF-2 treatment results in the inhibition of the differentiation of oligodendrocyte lineage cells to become mature oligodendrocytes [53,54]. However, the differentiation stages of the cells were different from those observed in vivo, as FGF-1 treatment in the spinal cord transection model resulted in less NG2+CNPase+ and NG2− CNPase+, whereas in vitro there was significantly more NG2+ CNPase + cells and NG2− CNPase+ cells.

### 3.6. Testing the Effect of FGF-1 on the Oligodendrocyte Lineage Using Blocker to the FGF-R

#### 3.6.1. Blocking of FGF Receptor Increased NG2+OX42− Cell Numbers in the Oligodendrogenesis Site

As there are many factors influencing the oligodendrocyte phenotypes during spinal cord injury or axon degeneration, we seek to determine whether FGF-1 is crucial for the regulation of the oligodendrocyte differentiation in our model. When PD173074, an inhibitor of FGF-receptor 1, was added to the treatment site in our spinal cord transection model with FGF-1, the inhibition of NG2+OX42 phenotypes was reversed (Figure 14).

#### 3.6.2. Blocking of FGF Receptor Failed to Bring back the APC+ Mature Oligodendrocyte Cell Numbers in the Area Nearest to the Trauma Site

When PD173074, an inhibitor of FGF-receptor 1, was added to the treatment site in our spinal cord transection model with FGF-1, the number of APC+ cell numbers nearest to the trauma site was even less than that treated with FGF-1 (Figure 15).

### 3.7. Verification of a Migration Plus Proliferation Theory: The NG2+ Cells in the Grey Matter and in the Back-Degeneative White Matter Track of Dorsal Funiculus at 8 mm

Movement or uneven distribution of OPC toward injury is a major factor that was absent from the cell culture model. OPC was known to migrate toward the injury epicenter in the acute stage [55]. From the above sections, we observed that the number of OPC or pre-myelinating oligodendrocytes was fewer in the FGF-1-treated rats at the site of active oligodendrocyte genesis. At the same time, there are more Nkx2.2+ OPC cells proliferating in the FGF-1-treated group. To explain this discrepancy, we tested the possibility that cell migration or discrepancy in cell distribution was responsible for the higher number of OPC cells in the transected group. We tested the NG2+ cell number in the site of interest (5 mm from the trauma site) and further away from the trauma site (8 mm from the trauma site). It is found that there are more NG2+ cells in the FGF-1-treated rats further away from the trauma site in the grey matter and in the dorsal funiculus (Figure 16). This demonstrated that the fewer number of OPC in the oligodendrocyte genesis site is due to uneven distribution caused by FGF-1 treatment.

## 4. Discussion

### 4.1. Oligodendrogenesis in Contusion Model vs. Transection Model

The dorsal column in the spinal cord consists mostly of ascending gracile fasciculus and cuneate fasciculus tract and only a small fraction is the descending cortical spinal tract (see Figure 15 for location of these tracts). We chose the caudal spinal dorsal tracts nearest to the transection site as it is a good site for examining the back-degenerative axons and associated demyelination. After a contusion injury, the cavity forms centering the grey matter of the spinal cord and extend to the periphery [5]; therefore, the demyelinated axons in the dorsal column are a mixture of the descending cortical spinal tract and ascending dorsal funiculus (Figure 17, left, contusion) [56]. Although the contusion model is more similar to the real-life injury scenario, the axons in the injury area were a mix of the Wallerian fragment and the back-degenerative fragment (Figure 17, left). These two types of axons differ in their subsequent survival and inflammatory degeneration, and hence their associated oligodendrocytes and OPCs might behave differently. In our complete transection model, on the other hand, the Wallerian degenerative tracts were separated from the residual, back-degenerative stump that was attached to the neuronal cell body by a 5 mm gap (Figure 17, right, complete transection). The back-degenerative gracile fasciculus and cuneate fasciculus could be anatomically distinct from the Wallerian degenerative cortical spinal tract in a complete transection model. Therefore, our complete transection model provided a window to observe oligodendrogenesis that are associated with a singular type of axon injury in the spinal cord.

Oligodendrocytes can differentiate by default, but they are also regulated by the signals from the axons. After SCI, how much OPC proliferation and differentiation contribute to axon integrity and subsequent functional stability for spared tissue remains debatable. Although white matter tissue preservation correlates to the preservation of good function, at least half of the individuals with complete loss of motor and sensory function below the injury (a.k.a. clinically complete) were anatomically incomplete [57]. It is postulated that spared but non-functional tissue could be a disease target i.e., remyelination to protect the axon might result in functional recovery, since only very few axons are needed (6% spared CST axons in humans). In short, remyelination may be a target for spinal cord injury therapy.

In this paper, we used a spinal cord injury model in which the spinal cord was completely transected and regeneration was attempted by grafting with peripheral nerves with a supplement of fibrin glue added with FGF-1. Previously, we have observed improvement in BBB score and functional recovery, which was recorded electrophysiologically [58], and regenerating axons were seen as early as 4 days post-operation in the grafted nerves (Figure 2 of this paper). This model provides a way to observe the OPC behavior during the acute stage in an environment that was known to be encouraging regeneration. Previously, we have observed back-labeled neurites that grow in the bordering area in the grey matter area and the white matter tract [23]. It is located in the periphery of the cavity and in the same area we observed CSPG in the transected model [44]. The back-degenerative tracts are more relevant than the Wallerian degenerative site concerning axon regeneration, as sprouting could happen in a back-degenerative tract at the distal site of the same axon, but not in the Wallerian degenerative tract. In sites where re-sprouting occurs, premature remyelination may be problematic and needs to be inhibited. This is a very different situation to consider than replenishing oligodendrocytes to protect a viable but chronic demyelinated axon. This is especially interesting when FGF-1, an OPC mitogen, is used for SCI repair, as it might induce premature myelination.

### 4.2. FGF-1 Reduced Number of Cells of Oligodendrocyte Lineage in SCI, while OPC Proliferation Was Increased

In our complete transected spinal cord, the number of OLIG2+ cells was more in the FGF-1-treated rats, which suggested that the proliferation rate was greater within the oligodendrocyte lineage in the FGF-1-treated model (Figure 3). When NG2(+) OX42(−) was used to detect the OPC, treatment with FGF-1 resulted in a reduced number of NG2(+) OX42(−) OPC (Figure 4). These NG2+ cells were bona fide OPC, as most of them were double stained with OLIG2 (Figure 5). When NG2(+) CNPase(+) and NG2(-)CNPase(+) were used to mark the less mature and more mature OPC in the oligodendrogenesis site, respectively, FGF-1 treatment resulted in the reduction in both CNPase+ NG2+ cells and CNPase+NG2− cells (Figure 6). APC+ mature oligodendrocytes were reduced in F4 and R4 when compared to T4 (Figure 7). R4 has less Nkx2.2−positive cells about 5 mm from the trauma center (Figure 8). The phenotypic profile of oligodendrocyte markers is summarized in Figure 18. Basically, the number of cells positive for oligodendrocyte lineage markers was lower in the FGF-1-treated rats in the area where oligodendrocyte generation in the untreated counterpart was most prominent. This contradicts our expectation of FGF-1 being a mitogen of the oligodendrocyte lineage.

Although the cells of oligodendrocyte lineages were less in FGF-1-treated rats, the proliferation rate of OPC was higher in the FGF-1-treated rats. The treatment of FGF-1 resulted in increased OPC numbers that were double stained of Nkx2.2 and BrdU with a 3-day repeated injection of BrdU (Figure 9, long pulse), while there is no significant difference when pulsed only 3 h before sacrifice (Figure 9, short pulse).

### 4.3. Addition of FGF-1 to the Spinal Cord Injury Site Resulted in Reduction of NG2+ OPC Population, and Reduced Demyelination

Normal spinal cord, when injected with FGF-2, the OPC starts to proliferate, and there was a demyelination observed. The myelin in the back-degenerative dorsal funiculus tract was compared for the T, F, and R groups (Figure 12). When we compared the markers of OPC and myelin near the oligodendrogenesis site between the (1) normal spinal cord + FGF-2, (2) completely transected, (3) transected + FGF-1, (4) transected + FGF-1 + nerve graft (5) mixed neuron-glia culture + FGF-1 (summarized in Table 2), it was concluded that FGF-1 treatment affect cells of the oligodendrocyte lineage and myelin different when FGF-1 was treated to the complete transected spinal cord, or the normal non-injured spinal cord.

Myelination and destruction of myelin take more time than the generation of oligodendrocytes. The main process during the time we are looking at is more of demyelination rather than remyelination, although it could be a mixture of both. This is confirmed by our own observation that there was more remyelination at the site nearer to trauma at 7 post-operation in R7. Demyelination is the loss of myelin which is associated with oligodendrocyte cell death during normal spinal cord injury. However, the death of oligodendrocytes was caused by local toxicity such as oxidative stress or excess glutamate, but demyelination is the work of macrophage phagocytosing. In situations where toxicity is not coupled to macrophage phagocytosis, it is not necessarily true. We have conducted previous experiments to demonstrate that the R and T models both have sufficient amounts of local oxidation for apoptosis to happen. However, we observed different macrophage populations for the T (mostly M1 macrophage) and R (mostly M2 macrophage) models. We argue that the reduced mature oligodendrocyte number was due to the same level of oligodendrocyte death plus less replenishment of mature cells for the FGF-1-treated model. For the preserved myelination, we think it was because of the differential macrophage type present (M1 is destructive and M2 protective).

### 4.4. A Delay in the Replenish of Oligodendrocytes

The APC+ cells near the trauma site were reduced in the FGF-1-treated model compared to the transected rats (Figure 7 also summarized in Figure 18). At the same time, the apoptosis rate detected by tunnel staining or staining of caspase-3 was not different between the two models (Figure 10 and Figure 11). The death of oligodendrocytes after spinal cord injury happened acutely within 3 days, and so is the appearance of APC+ oligodendrocytes in the oligodendrogenesis sites. The fact that there was no significant difference when treated with FGF-1 after short pulsed for Nkx2.2+BrdU cells, while long pulsed BrdU yield significant more Nkx2.2+BrdU+ cells, suggested the OPC underwent proliferation soon after injury but not so much at the end of the 3 day BrdU labeling period. This indicated that by the 4th day, with FGF-1 treatment, the OPC cells proliferated and started to exit the cell cycle to become pre-myelinating oligodendrocytes. This is crucial as the premature expression of myelin protein could hinder axon regeneration and FGF has prevented the progression into this stage.

### 4.5. The Paradox Could Be a Results of a Combination of OPC Migration and Proliferation

If we considered there was a significant amount of OPC movement toward injury in our system, it could explain the paradox of increased OPC proliferation rate but less OPC number in FGF-1-treated rats, as well as the difference between in vitro and in vivo data. The rationale and additional supporting data are as follows.

#### 4.5.1. The In Vitro Data Was Different When Compared to the In Vivo Models

To further dissect the effect of FGF-1, we compared the action of FGF-1 to that of FGF-2 and found they act similarly to the generation and differentiation of OPC in vitro in a mixed culture of neuron glia, although FGF-2 was more potent than FGF-1 (Figure 13). In this experiment, NG2 and CNPase were double stained and NG2(+)CNpase(+) and NG2(−)CNPase(+) populations, representing the less mature and more mature OPC in the oligodendrogenesis site, respectively. FGF-1 treatment results in less CNPase+ OPC cells entering the more mature (CNPase+NG2−) stage. This observation is different from its in vivo model, in which CNPase+NG2− in F4 was reduced compared to T4 (Figure 6). This discrepancy could be interpreted by the fact that the OPC we saw in the transected rats may initially migrate from neighboring areas due to injury, while OPC migration did not significantly contribute to those observed in FGF-1-treated rats in the injury site (summarized and illustrated in Figure 19a). In the cell culture, the starting OPC cell number for the FGF-treated and non-treated cultures is similar; therefore, the cell number was only affected by the presence of FGF-1 and therefore was more in the FGF-1-treated culture (Figure 19b).

#### 4.5.2. Blocking of FGF Receptor Increased NG2+OX42- Cell Numbers in the Oligodendrogenesis Site while Failed to Bring Back the APC+ Mature Oligodendrocytes Number Back

To further test the hypothesis that the OPC distribution was the result of the combination of OPC migration and proliferation, we tested with a blocker to the FGF receptor. This block of FGF-1 action resulted in increased NG2+OX42− cells for FGF-1 + blocker rats (Figure 14), demonstrating that FGF-1 treatment directly resulted in the reduction in OPC numbers in FGF-1-treated rats. The block of FGF-1 action also resulted in a further reduction in mature oligodendrocytes than those with FGF-1 (Figure 15, APC cells). As the FGF-1 acts both as an OPC mitogen and inhibits OPC migration, the lack of its action resulted in OPC migration and accumulation in the injury site, but a lack of OPC proliferation and hence no further maturation.

#### 4.5.3. Comparison to Distal Site in Spinal Cord: There Was Less OPC Cells in the Distal Site in Transected Rats

To further test the hypothesis that the OPC distribution was the result of the combination of OPC migration and proliferation, we compared the NG2+ OPC numbers at 8 mm away from the trauma site between the T and F groups (Figure 16). The dorsal horn area (grey matter) and the dorsal funiculus (white matter) were compared. FGF-1 treatment results in more NG2+ cells. There was a basic level of NG2+ OPC residing in the spinal cord under normal conditions; therefore, the low number of OPC in this area in the T group represents a situation where the OPC migrate into the neighboring trauma site. This observation agrees with a hypothesis that the lower number in the injury site in FGF-1-treated rats is a result of a lack of migration into this area rather than a lack of proliferation of OPC cells. In the injury site of the FGF-1-treated rats, the OPC was lower to start with; therefore, although we observed proliferation within the OPC population (Figure 9), the total OPC number tested using the CNPase/NG2 or Nkx2.2 staining was both lower than the T group.

### 4.6. FGF-1 and FGF-2 Were Known to Inhibit Oligodendrocyte-Differentiation by Regulating Reversion of Mature Oligodendrocyte to Become OPC

FGF-2 (also known as basic FGF, bFGF) is a member of the FGF family. The action of heparan sulfate-degrading enzymes activates FGF-2 and mediates the process of angiogenesis during both wound healing of normal tissues and tumor development. FGF-2 is necessary for human embryonic stem cells to remain in an undifferentiated state in vitro [59]. In conjunction with BMP4, FGF-2 promotes the differentiation of stem cells into mesodermal lineages [60]. In the CNS, FGF-2 regulates the generated oligodendrocyte progenitor cells (OPC) from the embryonic tissue [22] and inhibits the differentiation of oligodendrocyte lineage cells (OLC) to mature oligodendrocytes [53,54]. Increased OL and remyelination was higher in FGF-2 −/− mice, indicating that endogenous FGF-2 inhibited oligodendrocyte differentiation in wild-type mice [53]. FGF-2 is a mitogenic factor for OPC [61,62]. FGF receptor 1 (FGFR1) signaling inhibited differentiation of OPC into mature oligodendrocytes, as demonstrated by in vitro cell model and by conditional knockout of FGFR-1 in a chronic demyelination mice model [54,63]. The action of FGF-1, another member of the FGF1, 2 sub-group, is relatively unknown. FGF-1 was reported to inhibit OPC differentiation similar to the action observed with FGF-2 in vitro [64]. It is known to activate all four types of FGF receptors [65]. It shows a neuroprotective effect after spinal injury [66], and has trophic effects on neurons from various CNS regions [67]. From these experiments, it was concluded that FGF1/2 have dual action on the OPC: they could be OPC mitogen, or they would halt OPC differentiation, depending on the condition of axons they were associated with, or the level of inflammatory process in the near vicinity.

### 4.7. FGF-1 Directly Control the Differentiation of OPC into Myelinating Oligodendrocytes

CNPase plays a critical role in the events leading up to myelination [51]. When FGF-1 is used in the spinal cord repair model, with the addition of an FGF-R blocker in the same treatment, the reversion of mature phenotype, or a delay of differentiation of OPC, was reversed (Figure 14 and Figure 15). Furthermore, when we compare the action of FGF-1 in vitro and in vivo, FGF-1 can reduce the NG2+CNPase− and NG2+CNpase+ in vivo (Figure 6) but cannot prevent the progression to the NG2+CNPase+ in vitro (Figure 13). This suggests that the combination of environmental factors and FGF-1 is sufficient for the delay of phenotypic progression and replenishment of oligodendrocytes.

### 4.8. Biphasal Effect of FGF-1 on the OL Lineage: Initial Promotion of OPC Proliferation, and Removal of FGF to Induce OL Differentiation

Our data demonstrated that FGF-1 promotes OPC proliferation in the acute stage, and possibly influences their migration. Our observation consolidated the presumably contradictory role of FGFs in different models in which although the acute proliferation of OPC promoted by FGF facilitates remyelination, the precise timely cessation of this event is needed for a successful remyelination. Further, our data demonstrated the superior effect of using FGF-1 in the spinal cord injury model. Firstly, it promotes the proliferation of OPC independently of macrophage reaction or demyelination. Secondly, unlike a transgenic approach, a non-permanent treatment of FGF allows the cessation of the FGF effect enabling the maturation of OPC to OL.

### 4.9. Benefit of Exogenous FGF-1 Lies Not in Promoting OPC Proliferation, but Delaying Entry into Myelinating Stage

The cross-sectional size of an axon alone may regulate myelination to a surprising degree. Neuronal activity that regulates myelination includes electronic conduction, vesicle release, glutamate, GABA, neuregulin, GDNF, PDGF-AA, etc. [68], and factors that activate the Notch pathway or the Wnt pathway [69]. The intrinsic program that is responsible for the OPC differentiation involves transcription factors that were responsible for OPC maturation including ASCL1 (MASH1), OLIG1, Nkx2.2, and SOX9, which are recruited as positive expression regulators. OLIG2 and SOX10 also have positive activity. Binding to upstream enhancers of the HES1 promoter acts as a negative regulator of transcription [70]. MyT1, Oct-6 (also called Tst-1/SCIP) and Brn-1/Brn-2 were also responsible for OPC differentiation [71].

Of these pathways, FGF-2 was known to upregulate PDGF-AA expression in OP and block their differentiation into oligodendrocytes [72,73]. Moreover, PDGF and FGF-2 can act in concert to increase glutamate receptor subunit expression [74]. Glutamate agonists, in turn, can modulate the phosphorylation of OP nuclear transcription factors such as CREB through a MAP-kinase-dependent signal transduction pathway [75].

After injury to the axons, without exogenous FGF, local cellular reaction already generates a certain level (if not enough) of OPC. Our data suggested that upon FGF-1 treatment, the OPC number was increased without entering into the myelinating stage. It also possibly acts by delaying the OPC migration to the injury site. Taking together, another piece of known fact is that excess generation of oligodendrocytes will die of apoptosis without support from appropriate axonal survival signal [11]; we should interpret the contribution of our exogenous FGF-1 not as promoting OPC proliferation but enabling the delayed entry into the myelinating stage which may help with the axonal regeneration and functional recovery during the SCI.

## 5. Conclusions

In summary, we studied the phenotypes in an oligodendrocyte genesis site at the acute stage of spinal cord injury. We demonstrated that the number of OPC and APC+ pre-myelinating oligodendrocytes was less in the F group than in the T group. Paradoxically, the mitotic OPC cells are more in the F group than in the T group. In embryonic spinal cord mixed culture, FGF treatment resulted in more OPC than non-FGF-1-treated culture, which is different from those observed in vivo. As migration of OPC toward injury is a major factor that was absent from the cell culture, and we found OPC distributed away from the injury in cells with FGF-1 treatment, we proposed that it was possibly a combination of migration and proliferation resulting in the reduction in the NG2+ OPC population at the oligodendrocyte genesis site when FGF-1 was added to the spinal cord injury in vivo.

## Figures and Tables

**Figure 1 biomedicines-10-02724-f001:**
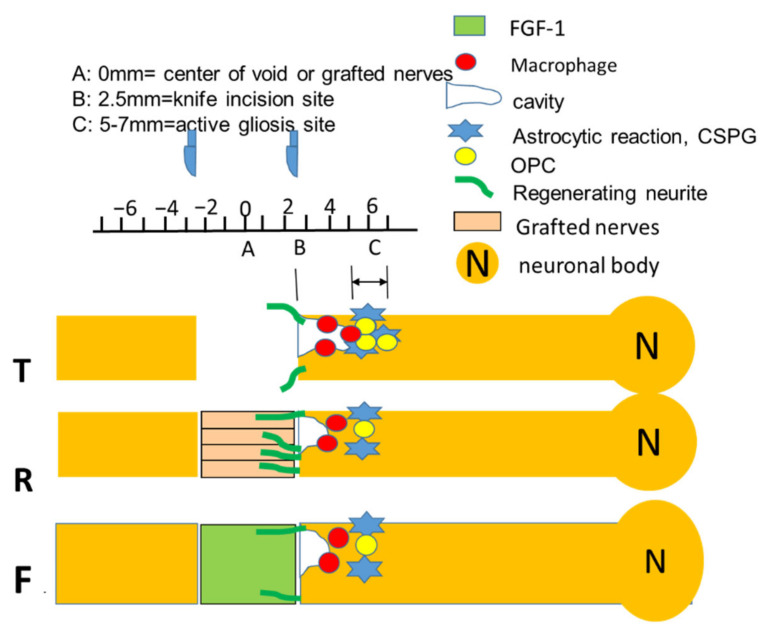
A diagram demonstrating the distance of incision site (B, marked 2.5) from the injury center (A, marked 0), as well as area that we previously consistently detect reactive gliosis and data from this paper collected (C, marked 5–7). (Details see Section 2).

**Figure 2 biomedicines-10-02724-f002:**
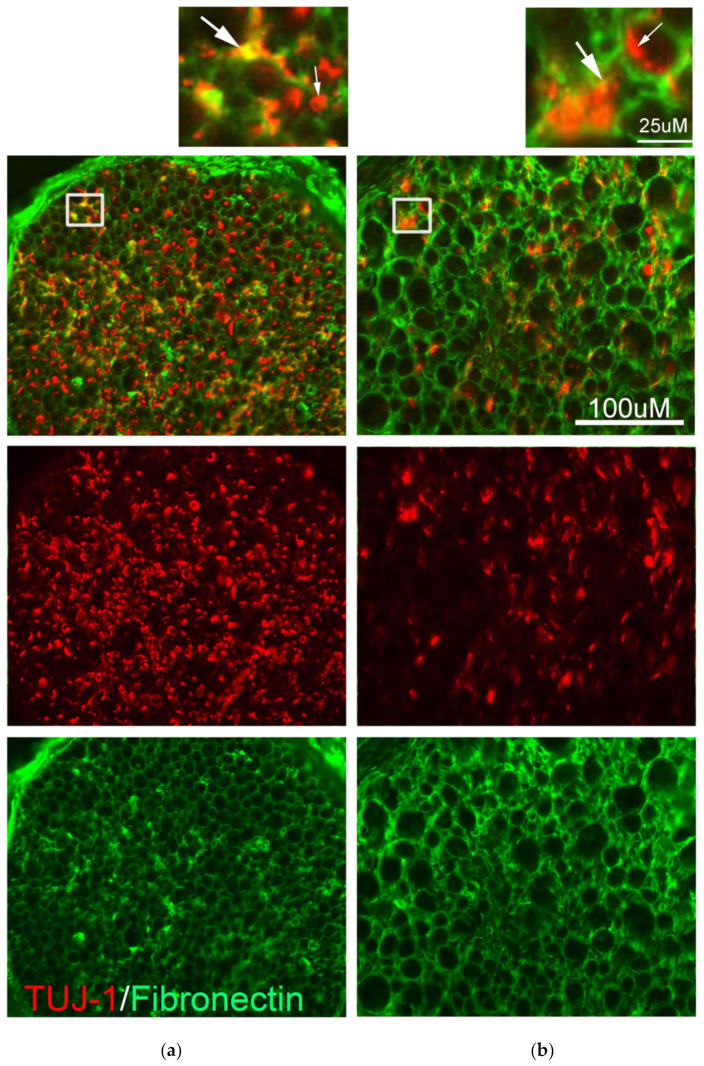
TUJ-1/fibronectin for axonal outgrowth. Representative figure of 3 experiments. Cross-sections were collected from the mid-way of the grafted nerves (marked A in the diagram from Figure 1). Antibody for fibronectin was used to stain the epineurium of the nerves, and TUJ-1 antibody stained both the remaining axons as well as the newly formed neurites. Insets were taken from the square in (**a**,**b**). The remaining axons of the nerve graft and the new neurites could be distinguished by their locations; the remaining, degenerating axons are encircled by the fibronectin-rich epineurium (small arrow), and the newly formed neurites grew on the fibronectin-rich epineurium (big arrow). (**a**) 4 days P.O. there are more denervated axons within the epineurium than those in (**b**) 14 days P.O.

**Figure 3 biomedicines-10-02724-f003:**
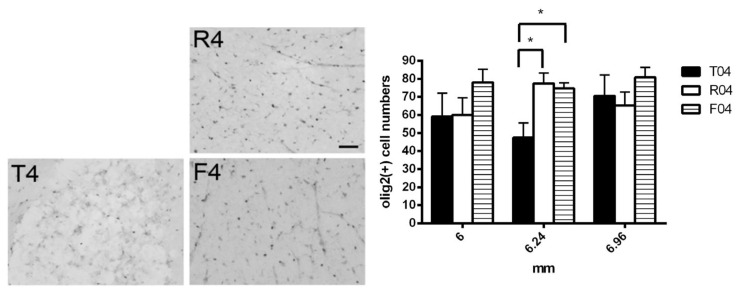
A cross-section near 6 mm from the trauma center showing the cells around the center cavity in the dorsal column. OLIG2 was used was a pan oligodendrocyte lineage cell marker. Statistical significance was evaluated using one-way ANOVA and Tukey’s test. Animal number used: T4 ((*n* = 7), R4 (*n* = 5), F4 (*n* = 6). * indicate *p* < 0.05. Scale bar =100 μm.

**Figure 4 biomedicines-10-02724-f004:**
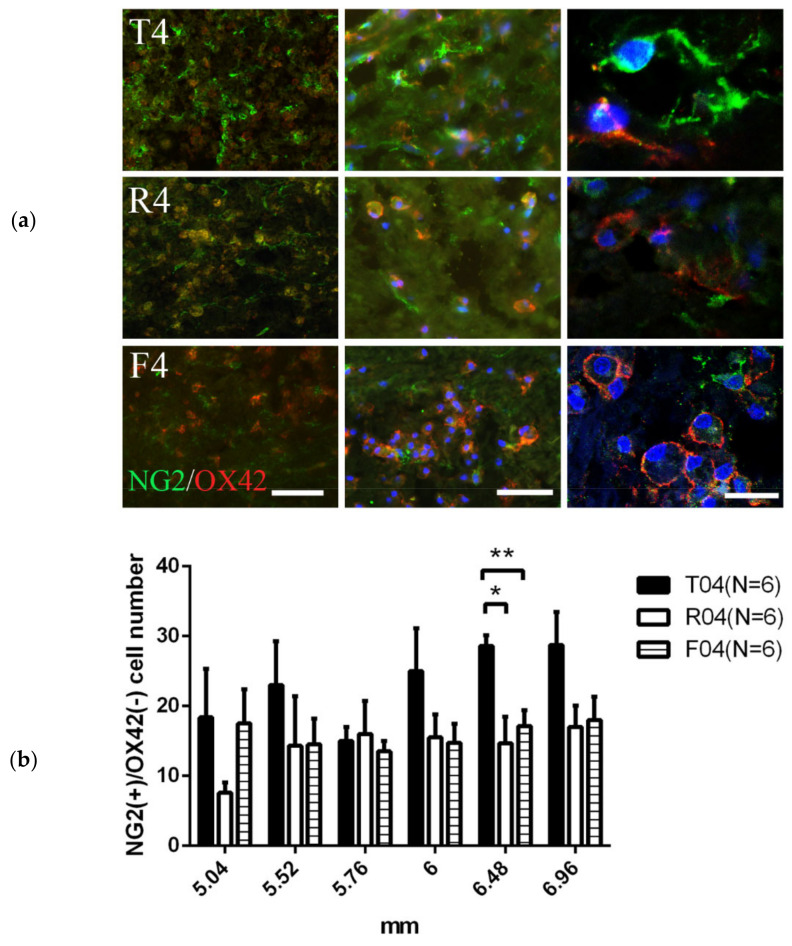
A cross-section showing the cells around the center cavity in the dorsal column of a caudal stump. (**a**) At 4 days post-operation, a degenerative cavity began to form in the center of the dorsal funiculus in the T4 group. NG2(+) Ox42(−) OPC were present near in the cavity site in T4 and to a lesser extent in F4 and R4. Animal number used: T4 (*n* = 6), R4 (*n* = 6), F4 (*n* = 6). (**b**) The numbers of NG2(+) Ox42 (−) OPCs were significantly different in some specific location around 6 mm from trauma center between T4 and R4, as well as between T4 and F4. Statistical significance was evaluated using one-way ANOVA and Tukey’s test. *: *p* < 0.05. **: *p* < 0.01. Data are presented as mean ± SEM. Animal number used: T4 (*n* = 6), R4 (*n* = 6), F4 (*n* = 6). Left scale bar= 100 μm, middle scale bar = 50 μm, right scale bar = 10 μm.

**Figure 5 biomedicines-10-02724-f005:**
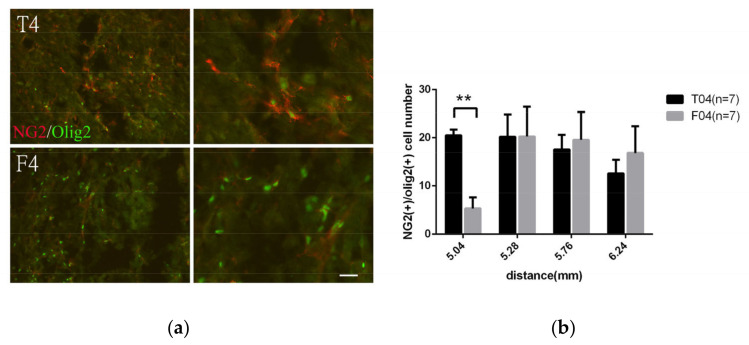
Cross-sections were collected in area ranging from 5 mm to 7 mm away from the trauma center. The sections were double stained withNG2 and OLIG2 to verify whether they were OPC cells (**a**). (**b**) The double-stained cells were counted in the injury site and compared between the T4 and F4 models. Pairwise *t*-test. **: *p* < 0.01. Data are presented as mean ± SEM. Scale bar = 50 μm.

**Figure 6 biomedicines-10-02724-f006:**
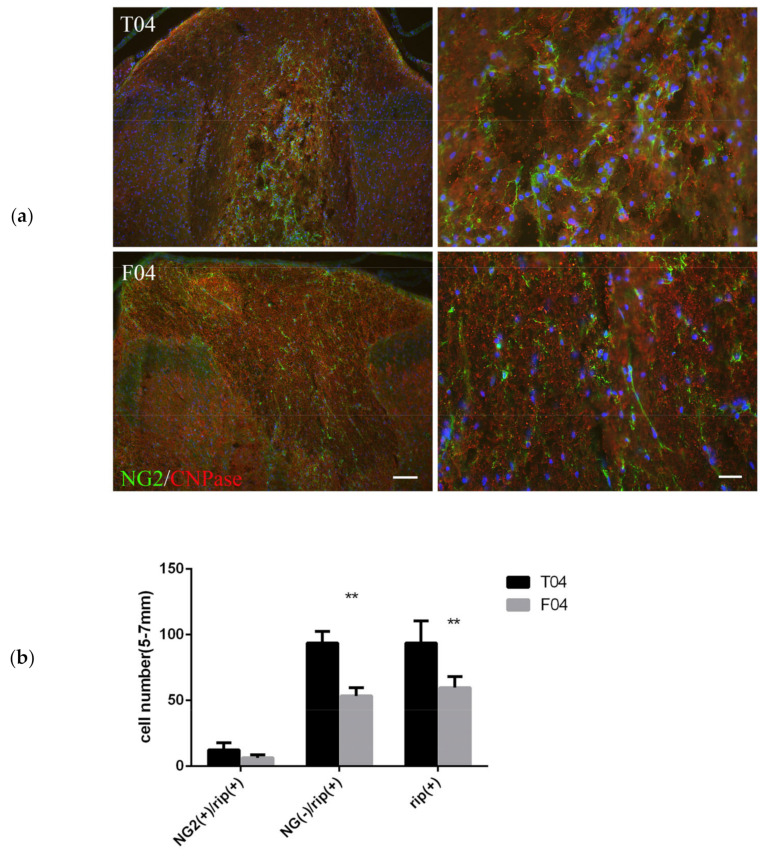
CNPase and NG2 double staining was used to study the maturation of the OPC cell populations. (**a**) Left panel is the overview of the dorsal funiculus. Right panel is the higher magnification of the cells on the periphery of the cavity, i.e., the oligodendrogenesis site. Animal number used: T4 (*n* = 5) F4 (*n* = 5). (**b**) Analysis of the two population NG2(+) RIP (+), NG2(−) RIP(+), and total RIP(+) cells. RIP is the antibody that stains CNPase antigen. Pairwise *t*-test. **: *p* < 0.01. Data are presented as mean ± SEM. Animal number used: T4 (*n* = 5) F4 (*n* = 5). Left scale bar = 100 μm, right scale bar = 200μm.

**Figure 7 biomedicines-10-02724-f007:**
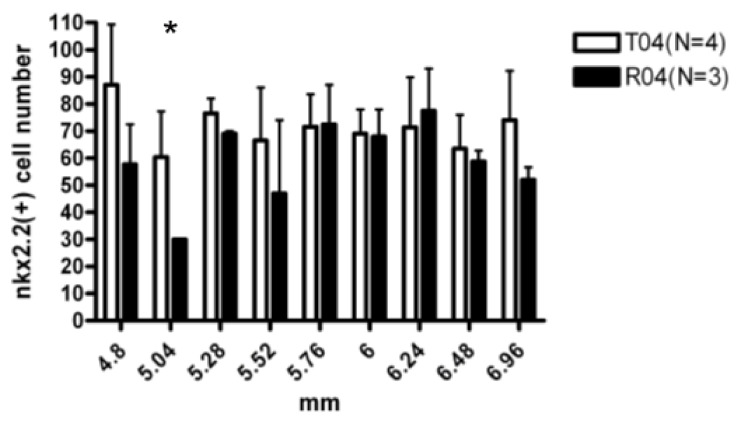
The Nkx2.2+ cells within the dorsal funiculus at 4.8 mm to nearly 7 mm. The total numbers of Nkx2.2 cell numbers were manually counted and are presented as the average amount within a set 200 pixel × 200 pixel area (1/6 of the total dorsal funiculus). The number of Nkx2.2+ cells present was less in the R4 than in T4 groups at 5 mm from the trauma center. Statistical significance was evaluated using one-way ANOVA and Tukey’s test *: *p* < 0.05. Animal number used: T4 (*n* = 4), R4 (*n* = 3).

**Figure 8 biomedicines-10-02724-f008:**
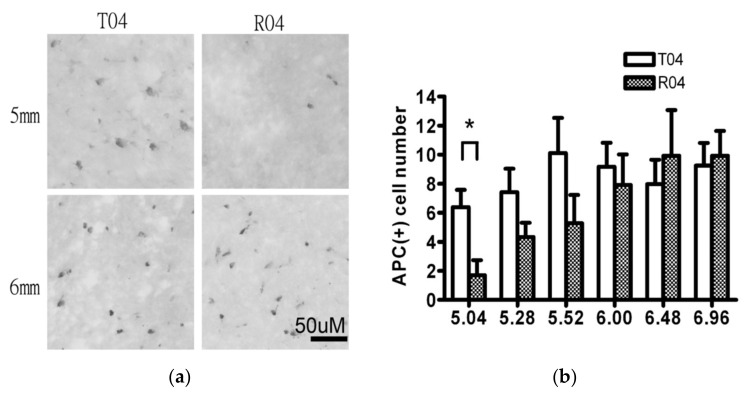
(**a**) The APC+ cells within the dorsal funiculus at 5 mm (upper panel) and 6 mm (lower panel). T4 (*n* = 6) R4 (*n* = 6). (**b**) The total numbers of APC+ oligodendrocyte numbers were manually counted and are presented as the average amount within a set 200 pixel × 200 pixel area (1/6 of the total dorsal funiculus). The number of APC+ cells present was not different between the R4 and T4 groups at 5.5–7 mm from the trauma center. However, adjacent to the open wound at approximately 5 mm from trauma center, the number of APC+ cells was significantly lower in the R4 group than the T4 group. Tested with pairwise *t*-test *: *p* < 0.05. Data are presented as mean ± SEM. Animal number used: T4 (*n* = 6) R4 (*n* = 6).

**Figure 9 biomedicines-10-02724-f009:**
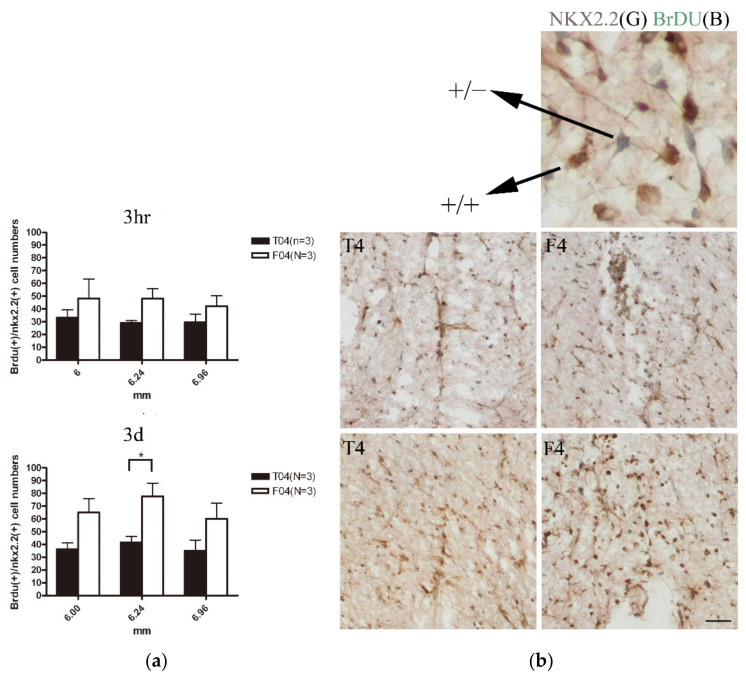
(**a**) Pairwise *t*-test. *: *p* < 0.05. Data are presented as mean ± SEM. Animal number used: T4 (*n* = 3) F4 (*n* = 3). (**b**) A cross-section near 6 mm from the trauma center showing the cells around the center cavity in the dorsal column. Nkx2.2 (+) was used to characterize OPC and pre-myelinating oligodendrocytes. For the short pulse, the rats were injected with BrdU 3 h before sacrifice. For the long pulse, the rats were injected with BrdU every day after operation and collected at 4 days post-operation. The sections were double stained with Nkx2.2 (grey) and BrdU (brown) with vectastain kit. Scale bar = 50 μm.

**Figure 10 biomedicines-10-02724-f010:**
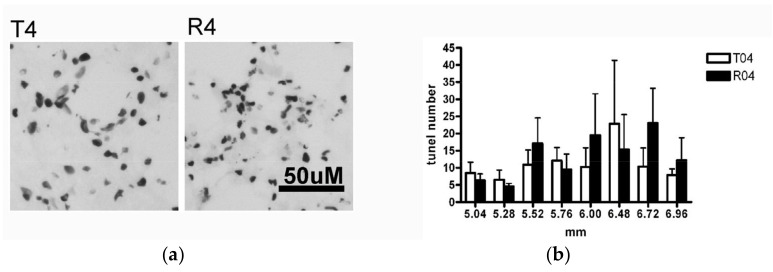
(**a**) The tunnel+ cells of the dorsal funiculus at 5.5 mm distance from the trauma center. This detects apoptotic cells in the dorsal funiculus, which were mostly oligodendrocytes. Animal number used: T4 (*n* = 6) R4 (*n* = 7). (**b**) At 4.8 mm to 6 mm from the trauma center, the number of tunnel+ cells was not different. Tested with pairwise *t*-test Animal number used: T4 (*n* = 6) R4 (*n* = 7). Scale bar = 50 μm.

**Figure 11 biomedicines-10-02724-f011:**
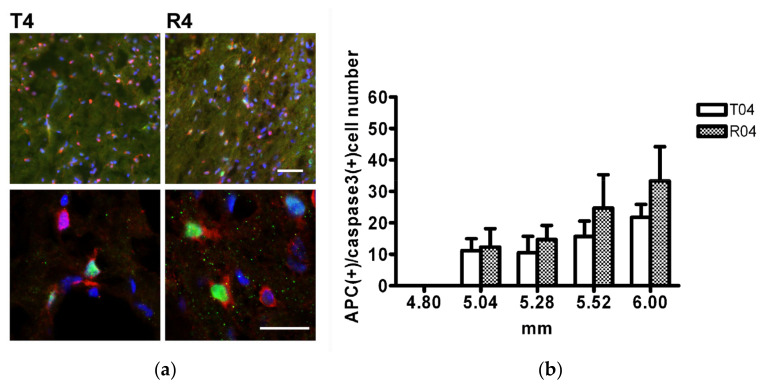
(**a**) The APC+ caspase+ cells of the dorsal funiculus at 5.5 mm distance from the trauma center. Double staining using APC and caspase-3 was used to detect apoptotic oligodendrocytes. Lower panel is a high magnification of the upper panel. Animal number used: T4 (*n* = 5) R4 (*n* = 3). (**b**) At 4.8 mm to 6 mm from the trauma center, the number of cells double stained with APC and caspase-3 was not different. Pairwise *t*-test. Animal number used: T4 (*n* = 5) R4 (*n* = 3). Upper scale bar = 50 μm. Lower scale bar = 10 μm.

**Figure 12 biomedicines-10-02724-f012:**
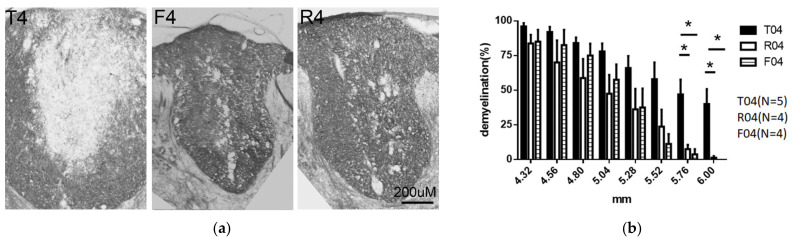
(**a**) Cross-sections showing myelin in the dorsal funiculus that was stained with Luxol fast blue at 5.5 mm to 5.75 from the trauma center. The myelin in supposedly proximal degenerative site in the R4 and F4 rats was almost intact; the degeneration found in the T4 rats was more profound. Animal number used: T4 (*n* = 5) R4 (*n* = 4) F4 (*n* = 4). (**b**) The proportion of demyelinated area to the total dorsal funiculus was manually judged and descriptive statistics were obtained. Luxol fast blue staining at 4 days P.O. in the transected rats demonstrated that the myelin was more destructed when compared to those in R4 and F4. Tested with pairwise *t*-test *: *p* < 0.05. Data are presented with SEM. Animal number used: T4 (*n* = 5) R4 (*n* = 4) F4 (*n* = 4).

**Figure 13 biomedicines-10-02724-f013:**
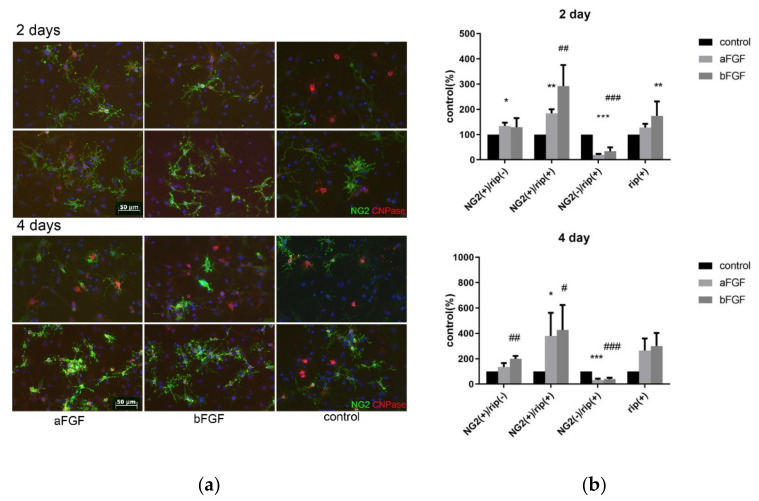
(**a**) The cells were cultured with FGF-1 (100 ng/mL) or FGF-2 (50 ng/mL), and were fixed and double stained with NG2 and CNPase at 2 and 4 days after FGF treatment. control (*n* = 4) aFGF (*n* = 4) bFGF (*n* = 4). (**b**) Number of 3 groups of cells was compared: NG2(+) CNPase(−), NG2(+) CNPase(+) and NG2 (−) CNPase (+). RIP was the antibody used to detect CNPase. Tested with Pairwise *t* test *: *p* < 0.05, **: *p* < 0.01, ***: *p* < 0.005. #: *p* < 0.05, ##: *p* < 0.01, ###: *p* < 0.005. Data are presented as mean ± SEM. Animal used: control (*n* = 4), aFGF (*n* = 4), bFGF (*n* = 4). Scale bar = 50 μm.

**Figure 14 biomedicines-10-02724-f014:**
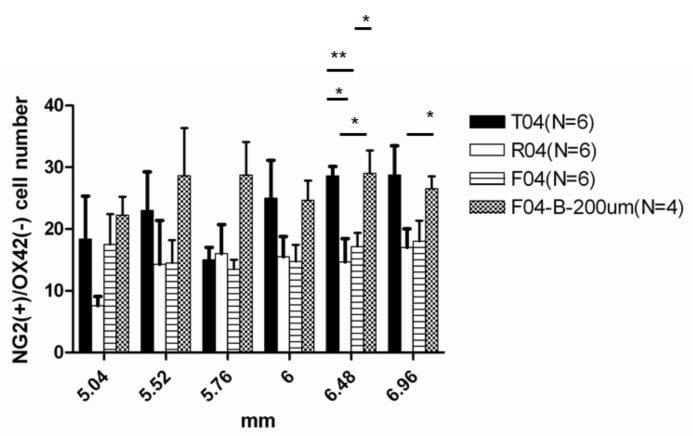
NG2+OX42−cells in the dorsal funiculus. Statistical analysis for the four groups: T4 = transected, R4 = PNS graft+ FGF-1, F4 = transected+ FGF-1, F4 + B = transected + FGF-1 + PD173074. At distances ranging from 6.68 mm to 6.96 mm from the trauma center, R4 and F4 rats showed lower NG2+OX42− cells compared to T4 rats, while F4+ blocker restored the number. Statistical significance was evaluated using one-way ANOVA and Tukey’s test *: *p* < 0.05, **: *p* < 0.01. Animal number used: T4 (*n* = 5) R4 (*n* = 6) F4 (*n* = 6) F4 + B (*n* = 4).

**Figure 15 biomedicines-10-02724-f015:**
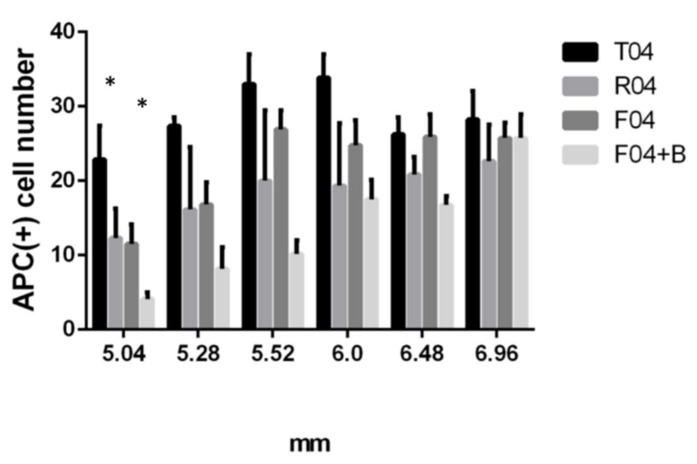
APC+ cells in the dorsal funiculus. Statistical analysis for the four groups: T4 = transected, R4 = PNS graft + FGF-1, F4 = transected+ FGF-1, F4 + B = transected + FGF-1+ PD173074. At distances ranging from 5.04 mm from the trauma center, R4 and F4 rats showed lower APC+ cells compared to T4 rats, while F4+ blocker further decrease the number. Statistical significance was evaluated using one-way ANOVA and Tukey’s test *: *p* < 0.05. Data are presented as mean ± SEM Animal number used: T4 (*n* = 4) R4 (*n* = 3) F4 (*n* = 8) F4 + B (*n* = 4).

**Figure 16 biomedicines-10-02724-f016:**
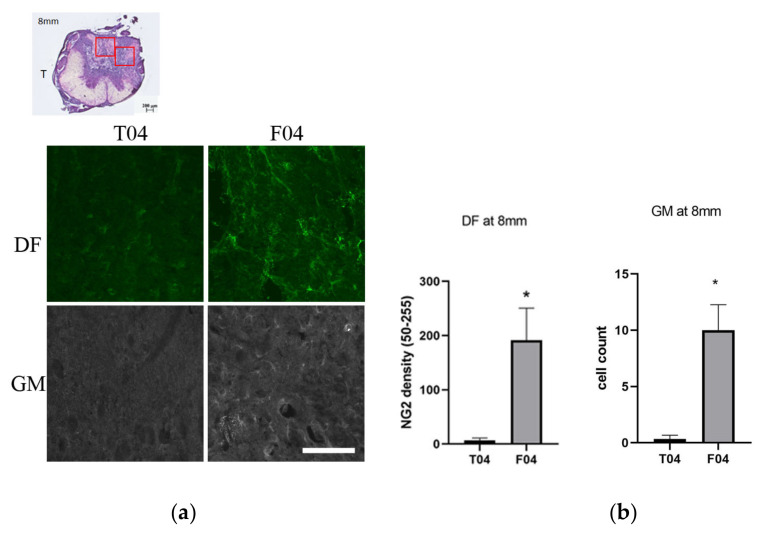
NG2+ cells in the grey matter (dorsal horn) and in the dorsal funiculus. (**a**) At distances ranging from 8 mm from the trauma center, F4 rats had more NG2+ cells compared to T4 rats, in the dorsal funiculus (DF and in the grey matter (GM). (**b**) Statistical analysis for the two groups: T4 = transected, F4 = transected + FGF-1, tested with pairwise *t*-test *: *p* < 0.05. Data are presented as mean ± SEM Animal number used: T4 (*n* = 3) R4 (*n* = 4). Scale bar = 100 μm.

**Figure 17 biomedicines-10-02724-f017:**
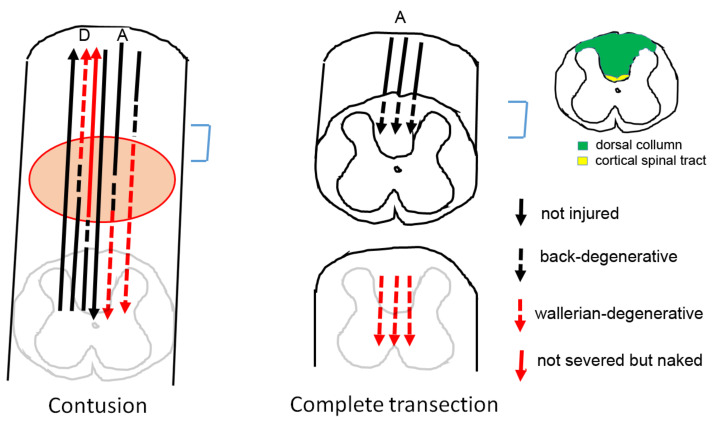
Comparison of white matter tract degeneration in contusion and complete transection model. D: descending tracks A: ascending tracks. In the caudal stump to the injury in the blue bracket area we saw a mixture of (left to right) descending not injured axon, descending Wallerian degenerative axon, descending not severed but naked axon, ascending no injured axon, and ascending back-degenerative axon. For the complete transection model, for the blue bracket area only ascending back-degenerative axon was observed.

**Figure 18 biomedicines-10-02724-f018:**
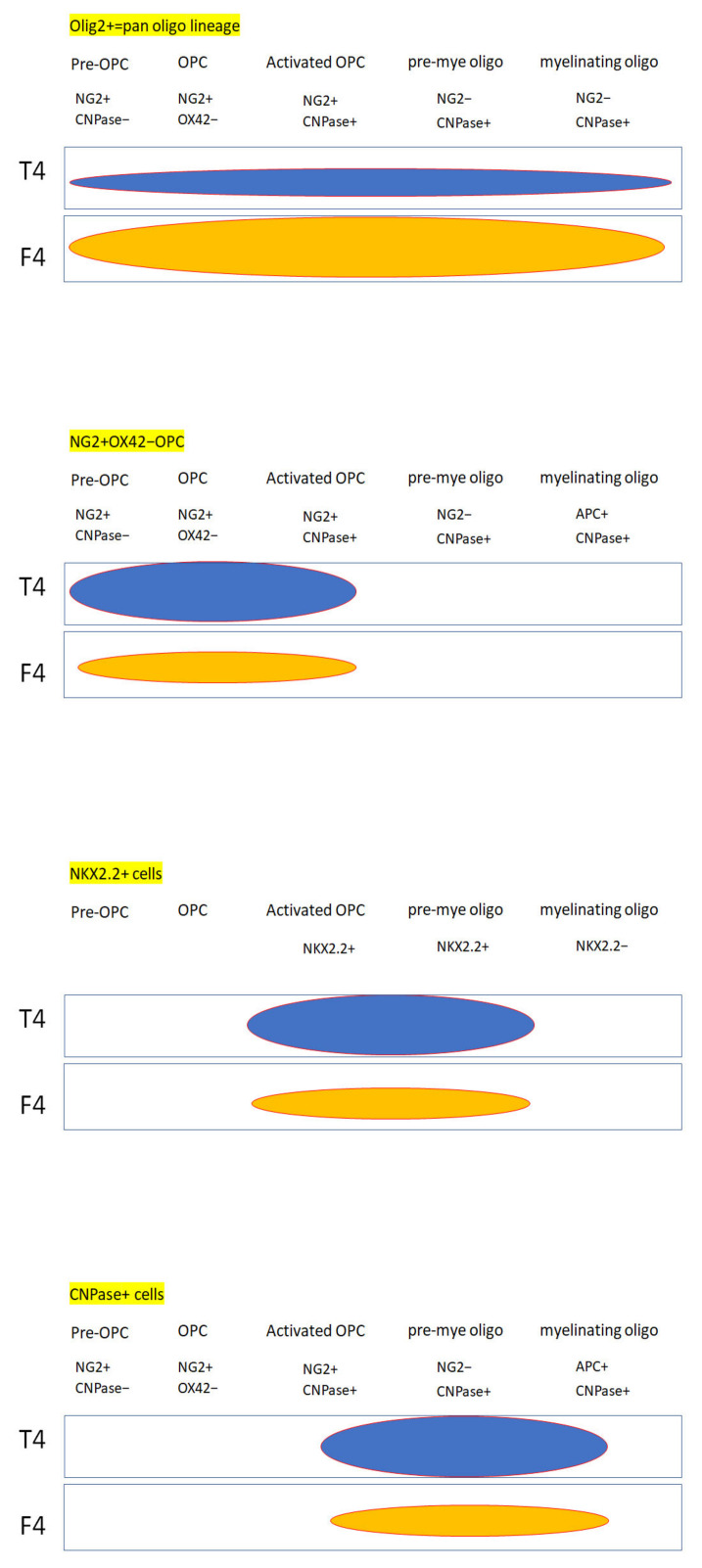
Comparison of differential markers expressed by the transected spinal cord and those treated with FGF-1.

**Figure 19 biomedicines-10-02724-f019:**
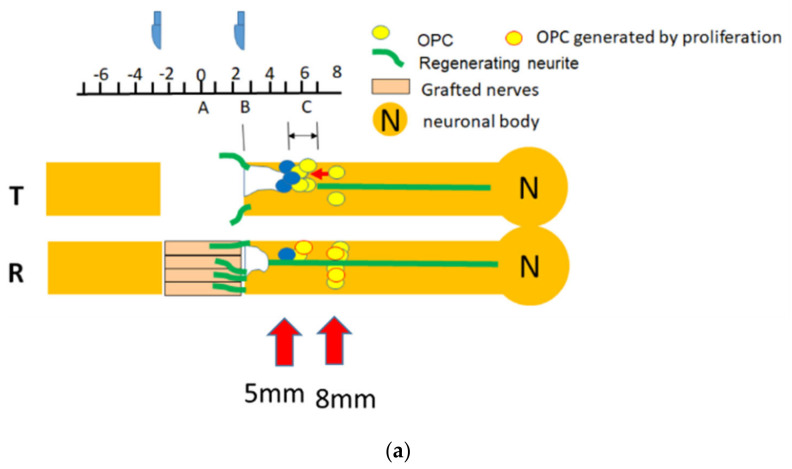
The OPC we saw in the transected rats may initially be migrated from neighboring area due to injury, while OPC migration did not significantly contribute to those observed in FGF-1-treated rats in the injury site (**a**). In the cell culture, the starting OPC cell number for the FGF-treated and non-treated cultures is similar; therefore, the cell number was only affected by the presence of FGF-1 and therefore was more in the FGF-1-treated culture (**b**).

**Table 1 biomedicines-10-02724-t001:** Molecular markers of oligodendrogenesis.

	Pre-OPC (Neural Stem Cell)	Non-Activated OPC	Activated OPC	Pre-Myelinating OL	Mature OL
OLIG2	+	+	+	+	+
NG2	?	+	+	−	−
Nkx2.2	−	−	+	+	−
CNPase	−	−	−	+	+

Simplified concept only: +: positive for the marker; −: negative for the marker; ?: controversial.

**Table 2 biomedicines-10-02724-t002:** Brief summary of different models of spinal cord injury treated with FGF-1 or 2.

	NG2 OPC	Myelin
Normal + FGF-2	U	D
T4	U	D
F4	D	same
R4	D	same
Culture + FGF-1 or 2	U	ND

U: up-regulated compare to normal, non-treated spinal cord; D: down-regulated compared to transected spinal cord; ND: not defined.

## Data Availability

All data is provided in full in the Section 3 of this paper.

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
