# Peer review of "Exogenous FGF-1 Differently Regulates Oligodendrocyte Replenishment in an SCI Repair Model and Cultured Cells"

_biomedicines, 2022, doi:10.3390/biomedicines10112724_

Round 1

Reviewer 1 Report

The authors shows the confirmation and the role of exogenous FGF-1 in SCI. Their proposal is supported by the results from a lot of experiments in vivo as well as in vitro. There have been some points for the authors consideration.

1) Title could be "Exogenous FGF-1 Differently Regulates the Oligodendrocyte Replenish  in a SCI Repair Model and Cultured Cells."

2) Figure 2. Description of experimental numbers in figure legends are required. For example, this panel is the representative of three experiments.

3) Figure 4a. Figure 2. Description of experimental numbers in figure legends are required.

4) Figure 6a. Description of experimental numbers in figure legends are required.

5) Figure 8a. Description of experimental numbers in figure legends are required.

6) Figures 10a and 11a. Figure 2. Description of experimental numbers in figure legends are required

7) Figure 12a, Figure 13a. Same as the above things.

8) Figure 16, In panels alphabetical letters are a-b. In contrast, in legends, A-B.

9) Table 1. ? is not suitable for Biomedicines.

10) Figure 18. It is very difficult to interpretate them as the summary. Should be simple and easy for understanding this study.

11) The same as the 10).

11)

Author Response

1) Title could be "Exogenous FGF-1 Differently Regulates the Oligodendrocyte Replenish in a SCI Repair Model and Cultured Cells."

A: changed as suggested.

2) Figure 2. Description of experimental numbers in figure legends are required. For example, this panel is the representative of three experiments.

A: added in the figure 2 legend: ‘Representative figure of 3 experiments”.

3) Figure 4a. Figure 2. Description of experimental numbers in figure legends are required.

A: added in figure 4 a legend ‘Animal number used: T4(n=6), R4(n=6), F4(n=6).’

4) Figure 6a. Description of experimental numbers in figure legends are required.

A: added in figure 6A legend ‘Animal number used: T4(n=5) F4(n=5).’

5) Figure 8a. Description of experimental numbers in figure legends are required.

A: added figure figure 8a ‘T4(n=6) R4(n=6).’

6) Figures 10a and 11a. Figure 2. Description of experimental numbers in figure legends are required

A: added in figure 10a ‘Animal number used: T4(n=6)R4(n=7)F4(n=4).’ In 11a added ‘Animal number used: T4(n=5) R4(n=3).’

7) Figure 12a, Figure 13a. Same as the above things.

A: Added in figure 12 a ‘ Animal number used: T4(n=5) R4(n=4) F4(n=4).’ 13a: ‘control (n=4) aFGF(n=4) bFGF(n=4).’

8) Figure 16, In panels alphabetical letters are a-b. In contrast, in legends, A-B.

A: changed as suggested.

9) Table 1. ? is not suitable for Biomedicines.

A: this author is not quite understand the question. Could the reviewer be more specific about the problem with the table? Is table not allowed?

10) Figure 18. It is very difficult to interpretate them as the summary. Should be simple and easy for understanding this study.

A: figure 18 is a graphic summary of the results presented in the paper. The points were: the NG2+OX42- OPC cells, NKX2.2+ OPC cells, NG2-CNPase+ premyelinating cells, populations in T group is more than the F group. in the text these points were stated in 4.3. FGF-1 reduced number of Cells of Oligodendrocyte Lineage in SCI, while OPC proliferation was increased.

This is followed by a paradoxical observation, in which the mitotic OPC is more in F group. this is explained by the following paragraph ‘Although the cells of oligodendrocyte lineages were less in FGF-1 treated rats, the proliferation rate of OPC was higher in the FGF-1 treated rats. The treatment of FGF-1 resulted in increased OPC numbers that were double stained of Nkx2.2 and brdU with a 3-day repeated injection of brdU (Figure 9, long pulse), while there is not significant difference when pulsed only 3 hours before sacrifice (Figure 9, short pulse).’

Hence the proposed model listed in figure 19, when the discrepancy between T and F in vivo model could be explained by a delayed migration in the F model.

11) The same as the 10).

A: please see answer to 10).

Reviewer 2 Report

The article is interestnig, however, some impovements are avised. 

I do not understand the title completely. It is not celarly written and something is missing. Can the authors rewrite the title or change it appropriately? 

The abstract is a little hard to understand. Can the authors rewrite it in a more pleasant way? 

In Methods, can authors explain whay the dorsal part of the spinal cord was used? Any advanatges in comaprison to anterolateral parts? 

Section 4.2 in Discussion, lines 559-579, on my opinion, is more suitable to be moved in the Intrduction. 

A Conclusion would be nice.

Author Response

I do not understand the title completely. It is not celarly written and something is missing. Can the authors rewrite the title or change it appropriately?

A: Title is changed to "Exogenous FGF-1 Differently Regulates the Oligodendrocyte Replenish in a SCI Repair Model and Cultured Cells."

The abstract is a little hard to understand. Can the authors rewrite it in a more pleasant way?

A: I have re-write and rid of some of the confusing points. Please see manuscript.

In Methods, can authors explain whay the dorsal part of the spinal cord was used? Any advanatges in comaprison to anterolateral parts?

A: the dorsal column was studied because it was an ascending tract and not mixed tract, therefore the back-degenerative oligodendrocyte genesis will not be mingled with the macrophage reaction of the Wallerian degeneration. The location of dorsal column is easily distinguished from the other tracks as it was delineated by the grey matter. This makes it a better tract than the anteriolateral tract. The latter, although also an ascending tract, is surrounded by other white matter tract of mixed ascending/descending direction, and could not be easily distinquished by simple observation of grey and white matter.

In section 4.1 in discussion, we wrote:

‘The dorsal column in the spinal cord consists mostly of ascending gracile fasciculus and cuneate fasciculus tract and only a small fraction is the descending cortical spinal tract (see Fig 15 for location of these tracts). We chose the caudal spinal dorsal tracts nearest to the transection site as it is a good site for examining the back-degenerative axons and associated demyelination.’

I think this paper is already very complicated, so to avoid further confusion I only answer reviewer’s questions here and did not incorporate my answer in the manuscript. If the reviewer wish so I will add to the text where it is suitable.

Section 4.2 in Discussion, lines 559-579, on my opinion, is more suitable to be moved in the Introduction.

A: thank you for the good suggestion. I have moved as recommended.

A Conclusion would be nice.

A: a conclusion is added at the end of the manuscript.

‘In summary, we studied the phenotypes in an oligodendrocyte genesis site at acute stage in spinal cord injur. We demonostrated that number of OPC and APC+ pre-myelinating oligodendrocytes were less in F group than T group. Paradoxically, the mitotic OPC cell are more in F group than in T group. In embryonic spinal cord mixed culture, FGF treatment resulted in more OPC than non-FGF-1 treated culture, which is different to those observed in vivo. As migration of OPC toward injury is a major factor that was absent from the cell culture, and we found OPC distributed away from the injury in cells with FGF-1 treatment, we proposed that it was possibly a combination of migration and proliferation resulted in reduction of NG2+ OPC population at oligodendrocyte genesis site when FGF-1 was added to the spinal cord injury in vivo.’